# Linking high GC content to the repair of double strand breaks in prokaryotic genomes

JL Weissman📵, William F. Fagan, Philip L. F. Johnson📵*

Department of Biology, University of Maryland - College Park, College Park, Maryland, United States of America

* plfj@umd.edu

## Abstract

Genomic GC content varies widely among microbes for reasons unknown. While mutation bias partially explains this variation, prokaryotes near-universally have a higher GC content than predicted solely by this bias. Debate surrounds the relative importance of the remaining explanations of selection versus biased gene conversion favoring GC alleles. Some environments (e.g. soils) are associated with a high genomic GC content of their inhabitants, which implies that either high GC content is a selective adaptation to particular habitats, or that certain habitats favor increased rates of gene conversion. Here, we report a novel association between the presence of the non-homologous end joining DNA double-strand break repair pathway and GC content; this observation suggests that DNA damage may be a fundamental driver of GC content, leading in part to the many environmental patterns observed to-date. We discuss potential mechanisms accounting for the observed association, and provide preliminary evidence that sites experiencing higher rates of double-strand breaks are under selection for increased GC content relative to the genomic background.

**Data Availability Statement:** All data used came from public repositories. Completely sequenced prokaryotic genomes were from NCBI's non-redundant RefSeq database (ftp://ftp.ncbi.nlm.nih.gov/genomes/refseq/). Relationships between

## Author summary

The overall nucleotide composition of an organism's genome varies greatly between species. Previous work has identified certain environmental factors (e.g., oxygen availability) associated with the relative number of GC bases as opposed to AT bases in the genomes of species. Many of these environments that are associated with high GC content are also associated with relatively high rates of DNA damage. We show that organisms possessing the non-homologous end-joining DNA repair pathway, which is one mechanism to repair DNA double-strand breaks, have an elevated GC content relative to expectation. We also show that certain sites on the genome that are particularly susceptible to double strand breaks have an elevated GC content. This leads us to suggest that an important underlying driver of variability in nucleotide composition across environments is the rate of DNA damage (specifically double-strand breaks) to which an organism living in each environment is exposed.

prokaryotes were from the SILVA Living Tree (https://www.arb-silva.de/projects/living-tree/). Clusters of related genomes were from the Alignable Tight Genomic Cluster (ATGC) database (http://dmk-brain.ecn.uiowa.edu/ATGC/). Prokaryotic trait data were from the ProTraits database (http://protraits.irb.hr/). Linkages between genomes and restriction enzymes were from the REBASE database (http://rebase.neb.com/rebase/rebase.html). Intermediate data files and code may be found at: https://github.com/jlw-ecoevo/gcku.

**Funding:** JLW was supported by a GAANN Fellowship from the U.S. Department of Education and the University of Maryland as well as a COMBINE Fellowship from the University of Maryland and funded by NSF DGE-1632976. WFF was partially supported the U.S. Army Research Laboratory and the U.S. Army Research Office under Grant W911NF-14-1-0490. PLFJ was supported in part by NIH R00 GM104158. The funders had no role in study design, data collection and analysis, decision to publish, or preparation of the manuscript.

**Competing interests:** The authors have declared that no competing interests exist.

## Introduction

Prokaryotic genomes vary widely in their GC content, from the small genomes of endosymbionts with low GC content (as low as 16% [1]) to the larger genomes of soil dwelling microbes with high GC content (> 60% [2, 3]). This bias in content might naturally be assumed to arise from biases in mutation rates, but a puzzle arose when observational studies surprisingly revealed a GC→AT mutational bias (which implies an expected equilibrium GC content < 50%) in genomes with actual GC content > 50% [4, 5]; more recently, controlled mutation accumulation experiments showed that even genomes with < 50% actual GC content still have greater GC content than expected from mutation rates [6]. This discrepancy between mutation rates and GC content implies that GC alleles fix at a higher rate than AT alleles. Two mechanisms could lead to biased fixation: selection directly on GC content [4, 5] or biased gene conversion (BGC), wherein homologous recombination favors GC alleles when resolving heteroduplex DNA mismatches [7]. Much debate has resulted over the relative contribution of these two mechanisms to the observed genomic GC content in prokaryotes [4, 5, 7–10].

This debate between proponents of the selection and BGC hypotheses continues, with many studies focusing on patterns of genetic diversity that by themselves cannot easily differentiate between these two hypotheses because recombination will also locally increase the efficiency of selection [9, 11]; however, the addition of phenotypic information provides the tantalizing clue that GC content correlates with shared environmental factors [2, 3, 12, 13] independent of phylogenetic similarity [3]. Thus, these environmental factors must either lead to an unknown selective advantage for high/low GC content [11] or lead to elevated rates of BGC through an as-yet unknown mechanism.

We noticed that many environments containing high GC content microbes, such as soils and aerobic environments [3, 12], induce relatively high rates of DNA damage in the form of double-strand breaks (DSB) that necessitate repair [14, 15]. For instance, in aerobes, this damage typically results from reactive oxygen species produced during metabolism [14] that can lead to DSBs by producing collapsed replication forks [16], as well as via a number of other mechanisms (often in conjunction with other stressors; [17–24]). In soil-dwelling microbes, DSBs are associated with desiccation and spore formation [25–28]. Even going back nearly 50 years, it was suggested that the rate of exposure to UV radiation, which can lead to DSBs [29], might be driving observed variation in genomic GC content among microbes [30].

To repair DSBs, microbes may use one of two pathways: homologous recombination (HR), or non-homologous end joining (NHEJ) [15]. HR machinery is ubiquitous across microbes [31], although it requires multiple genome copies to function. To-date, much work on GC content has focused on associating rates of HR locally along a genome (inferred using polymorphism data) with local GC content, which would be taken as evidence for the action of BGC [7]. We might also expect that organisms experiencing many DSBs would have an high overall recombination rate in order to repair these breaks. However, average rates of recombination in different genomes do not seem to be correlated with genomic GC content [5], despite the systematic environmental variation in GC content discussed above. It is possible that analyses that correlate global recombination rates and GC content looking across many genomes are too coarse-grained to reveal subtle differences between microbes leading to larger divergence in GC content over evolutionary time. Additionally, it is difficult to get accurate estimates of recombination rates from population-level polymorphism data, and it is unclear how strongly these rates would correlate with DSB formation specifically. Thus, an alternative, complementary indicator of high rates of DSB formation would be useful.

In contrast to HR, the NHEJ repair pathway is rarer and generally found in organisms experiencing DSBs with only a single copy of the genome present in the cell (e.g. during an extended stationary phase; [26]). Notably, we expect NHEJ repair to be favored only when HR is not an option, as NHEJ is generally considered a highly error prone pathway [15, 32]. NHEJ repair requires the presence of the highly conserved Ku protein [33, 34], which makes Ku presence/absence a useful indicator of genomes more/less likely to be subjected to high rates of DSBs during especially vulnerable periods (i.e., one genome copy present). We leverage Ku as an indicator of the rate of DSB formation and examine how the incidence of the NHEJ pathway co-varies with genomic GC content. We find a strong association between Ku presence and elevated GC content, and go on to discuss several mechanisms that could explain this pattern under a selection or a BGC paradigm.

## Results and discussion

### NHEJ and high GC content found in similar environments

A number of ecological factors have been associated with GC content in previous works, including aerobicity [12], nitrogen fixation [35], exposure to UV radiation [30], and growth temperature (although this last association has been disputed; [25, 36–38]). Notably, many of these associations are weak, and in general there is no known universal driver or mechanism that explains the high genomic GC content seen across many environments. We noted that many of the environmental factors correlated with GC content that have been identified in previous analyses are also associated with high rates of DNA damage, specifically DSBs. Perhaps, then, the unifying driver of GC content is the rate of DSB formation, and the environmental trends observed to-date can be attributed to this underlying driver.

While NHEJ presence is an imperfect indicator of DSB incidence in general, we expect this pathway to be especially common among organisms experiencing many DSBs during periods of slow or no growth [26]. Using a large-scale microbial trait database [39] paired with genomes from RefSeq [40], we identified known ecological correlates of NHEJ incidence as well as genomic GC content (some of which were redundant, S1 Fig). A principal component analysis of these traits revealed similar patterns of NHEJ incidence and high genomic GC content in trait-space (Fig 1), consistent with the idea that DNA damage is associated with genomic GC content. In fact, the pairwise correlation between an ecological trait and genomic GC content tracks almost perfectly with correlation between each trait and Ku incidence (S2 Fig).

Nevertheless, the inclusion of Ku along with ecological traits in a linear model to explain genomic GC content resulted in most other environmental traits still being statistically significant (S1 Table), indicating that either there is some aspect of the environment affecting GC content that is not attributable to DSBs or that NHEJ is an imperfect indicator of the rate of DSB formation (or both). In fact this is trivially true, as Ku presence is a discrete, binary variable whereas the rate of DSB formation is continuous. Despite the fact that Ku is not be the sole predictor of genomic GC content, the shared region of trait-space between NHEJ-capable organisms and high GC content organisms is quite striking (Fig 1).

### Organisms with NHEJ machinery have high GC content

Next, we looked directly at the Ku versus GC content relationship. Using a large set of genomes from RefSeq we found that genomes with Ku have a dramatically shifted GC content relative to genomes without Ku (Fig 2A, S3 Fig; Pearson correlation between GC content and Ku across genomes, $r = 0.54$, $p < 2.2 \times 10^{-16}$), even though Ku presence/absence is sprinkled throughout the prokaryotic phylogeny (Fig 2B). Indeed, this association remains highly significant even after formally correcting for phylogeny using phylogenetic regression with a

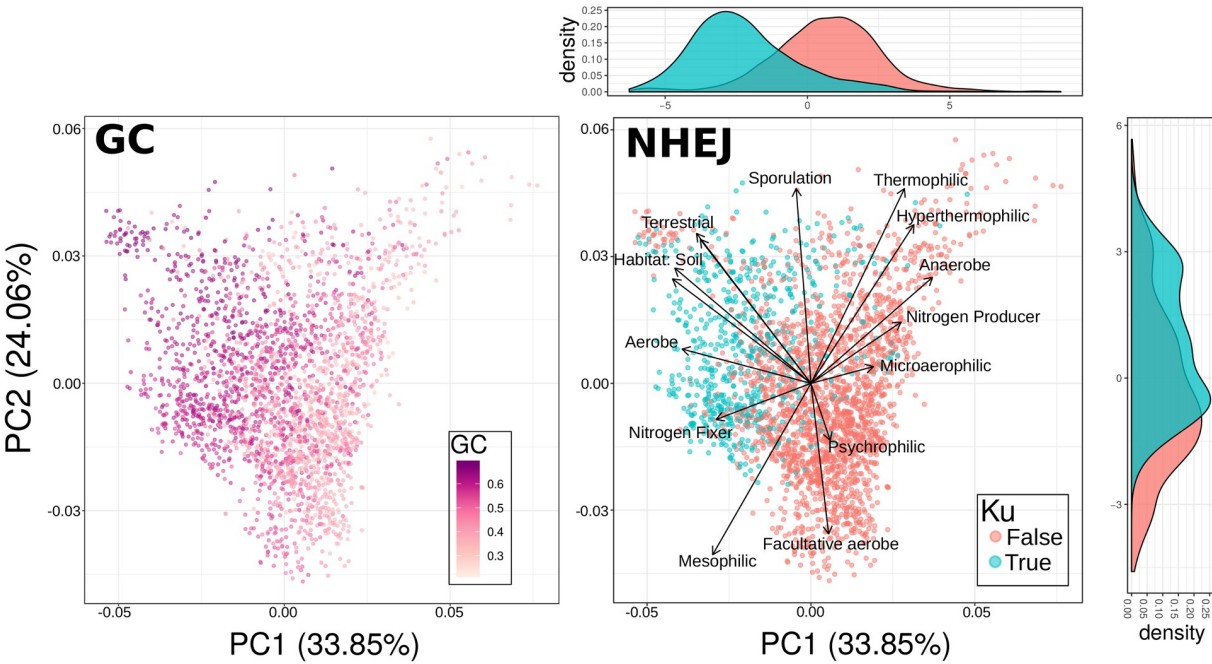

**Fig 1. Ku and high GC content share a particular region of trait space.** PCA of microbial trait data for select traits with species colored based on either their mean genomic GC content or whether they have known members that encode the Ku protein. Trait loadings signified by arrows. Note the clear separation of Ku and no-Ku organisms in trait space.

**(a)**    **(b)**

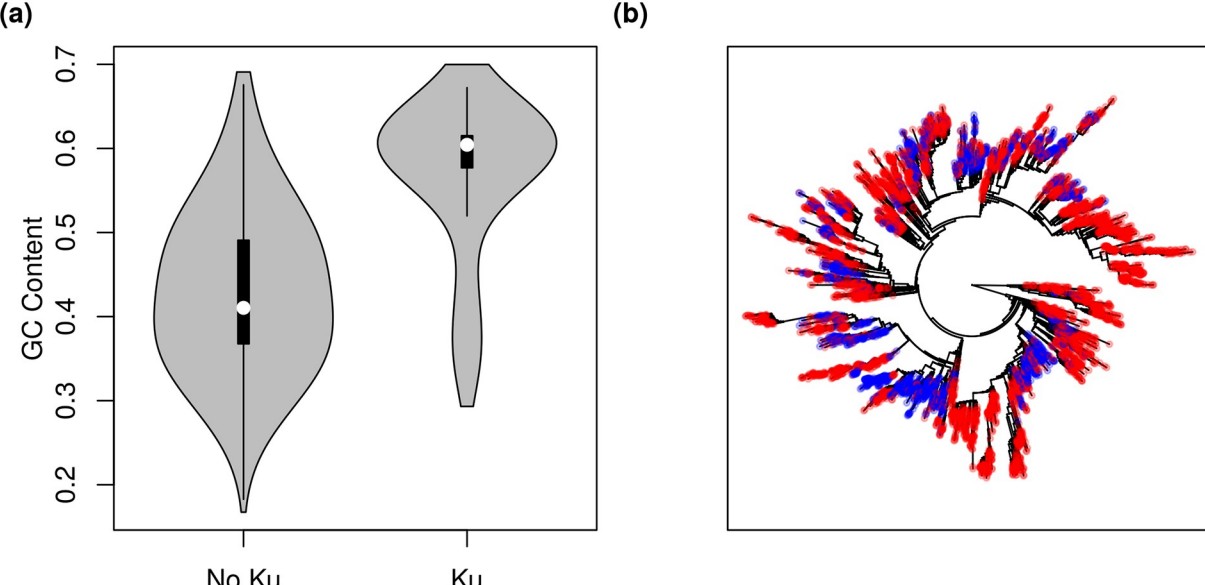

**Fig 2. The relationship between genomic GC content and the NHEJ pathway in prokaryotes.** (a) Microbes that code for the Ku protein tend to have much higher genomic GC content than those that do not (all RefSeq assemblies shown, 21389 out of 104297 genomes encode Ku). (b) Ku incidence mapped onto the SILVA Living Tree [64]. While Ku incidence is not randomly distributed across the prokaryotic tree, neither is it isolated to a particular clade. Organisms coding for Ku shown in blue, those not coding for Ku shown in red.

**Table 1. Coefficients and *p*-values for all phylogenetic regressions performed.** All tests significant after Benjamini-Hochberg correction ($\alpha = 0.05$). "Uniform Ku" refers to the dataset restricted to where Ku is always present/absent within each genus (see Methods).

| Data | Model | AIC | $\beta_{Ku}$ | $p_{Ku}$ | $\beta_{GenomeLength}$ | $p_{GenomeLength}$ | $\beta_{Interaction}$ | $p_{Interaction}$ |
|---|---|---|---|---|---|---|---|---|
| All | BM | -10754 | 1.29 | $1.465 \times 10^{-13}$ | 0.251 | $2.2 \times 10^{-16}$ | -0.191 | $2.522 \times 10^{-13}$ |
| All | OU | -10760 | 1.28 | $1.938 \times 10^{-13}$ | 0.248 | $2.2 \times 10^{-16}$ | -0.190 | $3.291 \times 10^{-13}$ |
| All (GC4) | BM | 6719 | 4.53 | $3.763 \times 10^{-12}$ | 0.812 | $2.2 \times 10^{-16}$ | -0.666 | $8.247 \times 10^{-12}$ |
| All (GC4) | OU | 6676 | 5.58 | $2.342 \times 10^{-12}$ | 0.827 | $2.2 \times 10^{-16}$ | -0.675 | $5.160 \times 10^{-12}$ |
| Actinobacteria | BM | -1997.131 | 3.26 | $2.2 \times 10^{-16}$ | 0.610 | $2.2 \times 10^{-16}$ | -0.483 | $2.2 \times 10^{-16}$ |
| Actinobacteria | OU | -2011 | 3.41 | $2.2 \times 10^{-16}$ | 0.644 | $2.2 \times 10^{-16}$ | -0.505 | $2.2 \times 10^{-16}$ |
| Firmicutes | BM | -1541 | 1.63 | 0.004541 | 0.104 | 0.028670 | -0.246 | 0.004358 |
| Firmicutes | OU | -1536 | 1.64 | 0.004242 | 0.110 | 0.022632 | -0.248 | 0.004088 |
| Proteobacteria | BM | -3181 | 0.719 | 0.01704 | 0.244 | $1.054 \times 10^{-11}$ | -0.105 | 0.01969 |
| Proteobacteria | OU | -3180 | 0.716 | 0.01744 | 0.238 | $3.377 \times 10^{-11}$ | -0.104 | 0.02011 |
| Uniform Ku | BM | -5167 | 2.22 | $6.517 \times 10^{-12}$ | 0.290 | $2.2 \times 10^{-16}$ | -0.325 | $2.032 \times 10^{-11}$ |
| Uniform Ku | OU | -5166 | 2.22 | $8.146 \times 10^{-12}$ | 0.287 | $2.2 \times 10^{-16}$ | -0.324 | $2.499 \times 10^{-11}$ |

Brownian motion (BM) model of trait evolution (Table 1). Our analysis is robust to the choice of evolutionary model, as repetition with an Ornstein-Uhlenbeck (OU) model of trait evolution yielded similar results (Table 1). Similarly, restriction of our analysis to a particular phylum (Actinobacteria, Firmicutes, and Proteobacteria, respectively; each with >1000 genomes on our tree) shows that this effect is not attributable to a single branch of the prokaryotic tree but is quite general (Table 1). Finally, to control for the possibility that Ku gain/loss via horizontal transfer is frequent and potentially confounding, we also restricted our analysis to a subset of the data where Ku presence/absence did not vary within each genera (discarding variable genera) and found qualitatively the same result (Table 1). In sum, the presence of NHEJ on a genome is positively associated with the GC content of that genome.

Importantly, we control for genome length in all our phylogenetic models, which potentially co-varies with Ku incidence and is known to be associated with genomic GC content in prokaryotes (Table 1 and S4 Fig). Interestingly, Ku presence and genome length have a significant negative interaction in their effect on GC content (Table 1).

Clearly organisms that encode Ku have a higher genomic GC content than organisms that do not, but can Ku help explain why organisms have a higher GC content than *expected*? Prokaryotes typically have higher GC content than predicted from their mutational biases, which are nearly always skewed towards AT alleles [4–6]. Does the observed association between NHEJ and genomic GC content contribute to this deviation? In other words, are GC alleles more likely to fix than the neutral expectation in Ku-encoding genomes, and is this deviance from neutrality larger than in genomes that do not encode Ku? Alternatively, it is possible that the error-prone NHEJ machinery simply leads to an excess of GC mutations. In this case, NHEJ incidence would help explain differences in mutational biases between microbes, but would not help explain the mystery of higher than expected genomic GC content among microbes.

Examining mutation accumulation experiments in detail, Ku shows no effect on GC↔AT mutational biases (S5 Fig; data from [6]). Since mutation accumulation data are limited, we also used the GC bias of inferred polymorphisms as a proxy for mutation. Similar to previous studies of prokaryotic GC content [4], and using the same intuition to that of the McDonald-Kreitman test for selection [41], we assume that polymorphisms within a population have experienced minimal selection and are therefore representative of the mutational biases of a given organism. Thus the GC content of polymorphisms within a population gives an estimate

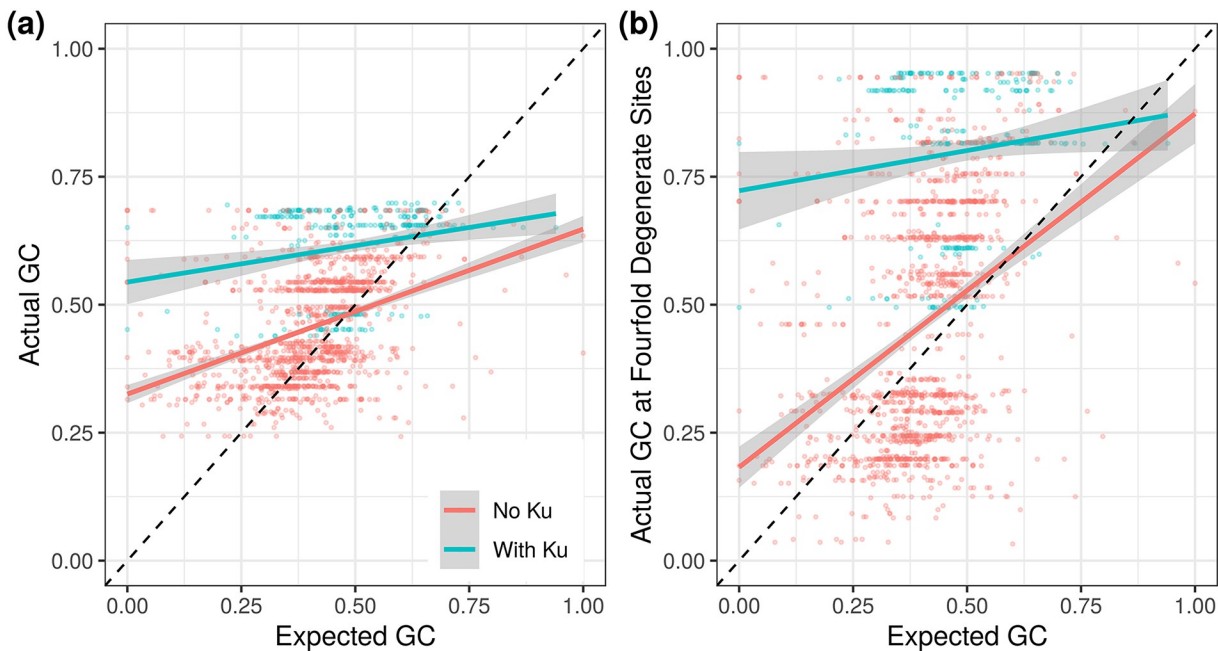

**Fig 3. Genomes with Ku appear to fix GC alleles at a greater rate than expected (either due to BGC or selection).** (a,b) Genomes with Ku have, on average, even greater elevation of GC content over expectation than genomes without Ku. Expected GC content was estimated from polymorphism data. This signal is conservative due to observed polymorphisms experiencing some effects of BGC/selection (see Methods for discussion).

of the GC↔AT mutational bias pre-selection, whereas the genomic GC content of an organism results from a combination of mutational biases and mechanisms that alter the probability of fixation of an allele, such as selection and BGC. We obtained multiple alignments of all orthologous genes for organisms in the ATGC database [42] belonging to clusters that contained at least three genomes (to identify and polarize polymorphisms) and used this dataset to compare the background GC content to the GC content of polymorphisms. We saw greater evidence for BGC/selection in genomes with Ku than without Ku, with the presence of Ku leading to higher observed GC content regardless of the expected GC content (Fig 3, S7 Fig). In order to minimize the effects of selection on our estimate of expected GC content, we repeated this analysis only using polymorphisms at fourfold degenerate sites and found qualitatively similar results, despite having only about a third as many informative polymorphisms (S6 and S7 Figs).

Thus, the association between Ku and genomic GC content is not due to differences in mutational bias. This implies that DSBs are either leading to selection for high GC content or influencing the rate and/or biases of homologous recombination to increase the overall action of BGC. We emphasize that this effectively rules out the possibility that biases during NHEJ repair are causing the observed patterns. NHEJ repair may be error-prone, but if those errors (i.e., mutations) were driving genome-wide GC-bias it would affect the GC-bias of polymorphisms as well as fixed alleles in the test described above.

Finally, we note that there is a small subset of genomes in Fig 2 that both encode Ku and have a low GC content ($< 40\%$). Of these, 80% belong to the family *Baccilaceae*. This family has uniformly low GC content ($> 99\%$ of genomes have GC content $< 50\%$, and 76% have GC content $< 40\%$), and an ancestral state reconstruction suggests that its most recent common ancestor encoded Ku (S8 Fig and see Methods), though Ku has been lost multiple times across

the group. We do not know why the *Baccilaceae* violate the pattern seen across the rest of the dataset; it may be an accident of evolutionary history or some particular aspect of this group's ecology and/or physiology.

## No apparent relationship between rates of homologous recombination and NHEJ

The above analyses suggest that GC alleles fix with a higher probability in organisms experiencing an elevated rate of DSB formation. If BGC were the primary driver of GC content evolution in prokaryotes, would we expect an association between damage and GC content as we see here? We can think of at least one plausible scenario. The formation of DSBs should stimulate recombination for repair, and assuming that recombination is biased we might expect rates of BGC to increase as the rate of DSB formation increases. We saw no positive association between Ku incidence and inferred rates of homologous recombination looking between genomes, as would be predicted by this hypothesis (S9 Fig with data from [43, 44], and S10 Fig with data from the ATGC database [42]). In fact the relationship appeared to be negative regardless of method to measure recombination rate (though not significant). That being said, the effects of BGC are typically only apparent locally, comparing GC content and recombination rates along a genome rather than between genomes [7]. The clearest evidence for BGC leading to high GC content in prokaryotes comes from Lassalle et al [7], who compared the GC content of genes that recombined frequently or rarely within genomes. We checked if any of the 21 taxa they studied typically encode Ku based on the genomes we downloaded from RefSeq above. Four taxa carry Ku at any appreciable frequency (>1%): *Mycobacterium tuberculosis* (100%), *Burkholderia pseudomallei* (100%), *Burkholderia cenocepacia* (87%), and *Bacillus anthracis* (68%). These organisms did not show a consistent association between recombination rate and GC content, unlike most other taxa in the study. *M. tuberculosis* and *B. pseudomallei* are highly clonal and did not present enough diversity for a complete analysis by Lassalle et al [7], though *B. pseudomallei* had a negative association between recombination rate and GC content at the third codon position (positive for GC content overall, in both cases not significant). *B. cenocepacia* presented enough data for analysis, but showed no relationship between recombination rate and GC content (again with a negative but non-significant effect at the third codon position). Finally, *B. anthracis* had inconsistent effects, with a significant negative association between recombination rate and GC content overall but a significant positive interaction when restricting to GC content at the third codon position. What to make of this? Among the very limited number of species that encode Ku in Lassalle et al.'s dataset, the evidence for BGC is not strong.

Given the small number of organisms in Lassalle et al.'s dataset that had Ku, we endeavoured to repeat this analysis using a larger set of organisms. Using the ATGC database (as we did with our analysis of polymorphism earlier), we obtained multiple alignments of all orthologous genes for each cluster of organisms [42]. We then classified genes as recombining or non-recombining using the PHI statistic [45]. Similar to Lassale et al. [7], we found that recombining genes had higher GC content than non-recombining genes, though this difference was small (paired *t*-test, $df = 154$, $p = 1.503 \times 10^{-11}$; Fig 4). Interestingly, while a link between recombination and GC content was apparent, it seemed to explain none of the difference between Ku-encoding and Ku-lacking organisms (Fig 4a). In fact the difference in GC content between recombining and non-recombining genes was actually smaller for Ku-encoding organisms than Ku-lacking ones, the opposite of what we would expect if recombination were driving the link between Ku and GC content (*t*-test, $df = 83.698$, $p = 0.0308$; Fig 4b).

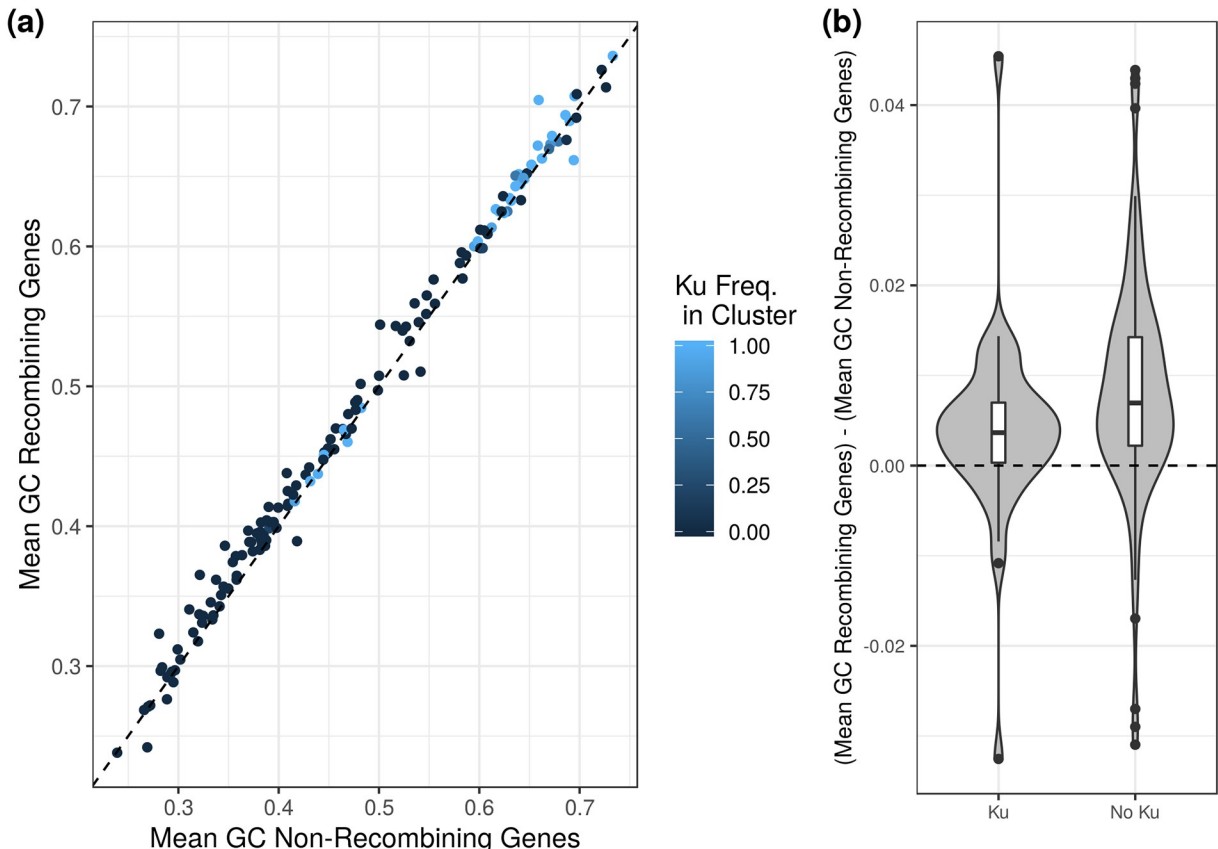

**Fig 4. Recombination contributes to GC content locally but cannot explain the relationship between GC content and Ku incidence.** (a) The mean GC content of genes with evidence for recombination (PHI statistic, [45], see Methods) plotted against the mean GC content of genes without evidence for recombination in a given closely related cluster of organisms (ATGC database [42]). Recombining genes have slightly higher GC content than non-recombining genes (points mostly lie above the dashed $x = y$ line). (b) The difference in GC content for recombining and non-recombining genes within a cluster is smaller for Ku-encoding than Ku-lacking clusters. Clusters classified as Ku-lacking if no members encoding Ku ($n = 114$) and Ku-encoding if at least one member has Ku ($n = 41$). Clusters excluded if no evidence for recombination was found for any of their genes.

Taking a step back, there are two primary reasons to disfavor BGC as the hypothetical mechanism underlying the observed positive association between NHEJ and genomic GC content. First, the extent to which BGC is a driver of genomic GC content in prokaryotes has been questioned. Studies using alternative methods to quantify recombination across the genome to those used presently by us and previously by Lassalle et al. [7] have found either no relationship between recombination rate and local GC content or inconsistent patterns between species (with some species even showing a negative relationship; [46, 47]). More recently, looking specifically at polymorphisms arising via recombination across many prokaryote species, Bobay and Ochman [10] reject the BGC hypothesis outright. They note that some positive relationship between recombination and GC content is apparent at a coarser scale (as seen here), but attribute this pattern to the increased efficiency of selection in regions of high recombination. Even in some microbial eukaryotes, where the evidence of BGC is thought to be strong, tetrad analysis has been unable to reveal any evidence of BGC leading to elevated GC content, including in genomes where recombination rate and GC content are locally correlated [48]. This suggests that correlative methods may be insufficient to conclusively demonstrate BGC,

and it has been suggested that high GC content may actually increase the rate of recombination locally (effectively reversing the logic behind the evidence for BGC; [49]).

Second, while organisms encoding Ku are likely to be experiencing DSBs, they are unlikely to experience high rates of recombination. NHEJ is thought of as an alternative pathway to HR, specifically used when HR cannot proceed because the genome is only present as a single copy [15, 26]. Thus we expect NHEJ to be favored specifically in situations where BGC is unlikely. While it is possible that high rates of damage could still favor both NHEJ and HR, albeit at different points in an organism's life cycle, the extremely strong and specific association between GC content and Ku suggests that this relationship may be particular to the specific conditions selecting for Ku (especially considering the absence of an association between recombination and GC content when looking between genomes [5]; S9 and S10 Figs). Nevertheless, we have insufficient information to completely rule out BGC as a mechanism at this time.

## High GC content near regions with frequent breaks

Given the inability of BGC to explain the association between NHEJ and high GC content (Fig 4), perhaps selection can provide an alternative hypothesis. Could it be that organisms with NHEJ machinery are under stronger selection for high GC content than those without? This leads to us to a puzzle: what fitness advantage might be conferred by GC content? In fact, high GC content may promote DNA repair, both by facilitating canonical NHEJ (i.e., Ku-dependent [50–52]) and alternative NHEJ (i.e., Ku-independent [32, 53]) pathways.

During DSB repair, the NHEJ machinery in prokaryotes takes advantage of homology in any short overhanging regions or nearby microhomology in order to help align the two broken DNA ends [50–52]. Any factor that stabilizes this interaction (e.g., high GC content via an increased number of hydrogen bonds) may have the potential to increase the efficiency of NHEJ repair [50–52]. It has also been shown that prokaryotes can employ alternative high-fidelity end-joining pathways that are independent of the NHEJ machinery [32, 53], and that these pathways are primarily dependent on short (2-5bp [32]) nearby microhomology (DNA ends are typically degraded to reveal internal homologies [32, 53, 54]) to tether the DNA ends together. It stands to reason that high GC content in these regions of microhomology might help stabilize the end-pairings and improve the efficiency of repair [32, 53]. In fact, in eukaryotes, high GC overhangs or microhomologies specifically promote the use of a similar NHEJ-independent, high-accuracy end-joining repair pathway [55, 56]. In these systems high GC content is thought to help tether the DNA together and thus perform a similar role to Ku [55, 56], though this has not yet been confirmed in prokaryotes. Nevertheless, this mechanism suggests that high GC content could help to ameliorate the negative effects of DSBs, especially in environments with high rates of DSBs but only a single genome copy. That being said, high genomic GC content alone cannot protect an entire genome, since most genomes will have at least some AT-rich regions. Thus a combined Ku and high GC content strategy would potentially be favorable for DSB-vulnerable microbes and could explain the strong positive relationship we observed between selection for high genomic GC content and the presence of Ku (Fig 2).

In addition, our hypothesis makes a novel testable prediction: regions of the genome that are especially prone to DSBs should be under selection to have higher GC content. Restriction modification (RM) systems provide an ideal test case as damage due to self-targeting is a known phenomenon (e.g. [57]), and we know the potential locations of self-targeting if we know the restriction enzyme recognition sequence. We hypothesized that restriction enzymes on a genome would be selected to target sites higher in GC content than expected from the

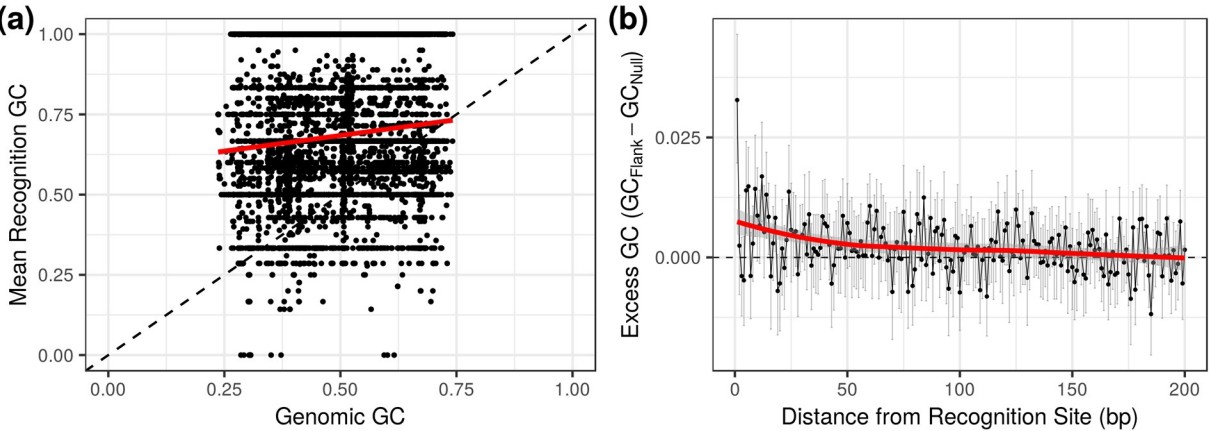

**Fig 5. Restriction sites are associated with elevated GC content.** (a) Restriction enzymes tend to target sequences with GC content higher than the genomic average. (b) Bases immediately flanking AT-rich restriction sites ($\geq$ 75% AT, $n$ = 214 genomes) have an elevated mean GC content. This signature mostly decays within 50bp of the recognition site. This pattern is particularly striking when looking at the first flanking base. Error bars represent bootstrapped 95% confidence intervals of the mean.

genomic background to help mitigate the effects of autoimmunity (otherwise they should match host background GC content because phage typically track their host nucleotide composition, often as a byproduct of optimizing codon usage bias for transcription in their host, e.g. [58]). We further predicted that, for restriction enzymes with low GC content recognition sequences, the bases flanking restriction sites on the genome would have elevated GC content. Both of these predictions were borne out. We analyzed the complete set of genomes and their listed restriction enzymes in the REBASE database [59] and found that restriction enzymes indeed tend to have higher GC content recognition sequences than their genomic background, and that the bases immediately flanking AT-rich recognition sites have elevated GC content (Fig 5). In principle, evidence of high GC content near breaks could also be taken as support for BGC (despite other evidence to the contrary [10, 48]) since the rate of HR repair should increase locally, meaning that ultimately experimental approaches will be needed to tease apart these hypotheses.

We emphasize that the idea that DSB formation selects for high GC content, while consistent with our data, is at this point largely speculative. By outlining this scenario we hope to enable experimentalists to design specific, mechanistic studies. Much of the debate over genomic GC content (including our present contribution), and especially its ecological role, has relied on population-scale correlative studies and has largely avoided mechanisms. Our "GC-tethering" hypothesis has the advantage of being amenable to laboratory-based investigations.

Finally, we caution that one is unlikely to see genome-wide differences in GC content when comparing across organisms with different numbers of restriction enzymes, since restriction sites comprise a very limited subset of loci along the genome (and self targeting should be somewhat restrained via methylation of the host chromosome). Presumably if self targeting was frequent enough to select for elevated GC content at a genome-wide scale, the corresponding cost of encoding these enzymes would be prohibitively high.

## Conclusions

We found a strong positive association between the presence of the NHEJ pathway on a genome and genomic GC content across prokaryotes. This association holds controlling for phylogeny and genome length and cannot be explained by mutational biases. The NHEJ repair

pathway is broadly but sparsely distributed across the prokaryotic tree (Fig 2), showing up in only about a quarter of genomes [33], and is expected to be favored among organisms experiencing high rates of DSB formation during periods of no or slow growth (i.e., only a single genome copy present so that HR is impossible; [15, 26]). This suggests that high GC content may be an adaptation to deal with DSBs when HR is not feasible, especially when rates of damage are high. In fact, we find that in regions of the genome where DSBs are likely to occur, GC content is locally elevated (Fig 5). Alternatively, the presence of NHEJ may be an indicator of high rates of DSBs in general, so that these organisms are also experiencing higher rates of HR for repair, and subsequently increased BGC. We discussed the relative merits of these two hypothetical mechanisms linking DSBs to genomic GC content, though at this point it is not possible to state conclusively which mechanism is the primary driver of the pattern we observe. It is also possible that some combination of BGC and selection is acting to increase genomic GC content in organisms experiencing DNA damage.

Regardless of the underlying mechanism, high risk of DSB formation is a common factor in many of the habitats that high GC content microbes have been shown to inhabit. While the presence of NHEJ cannot single-handedly explain high GC content in all organisms (there are many organisms incapable of NHEJ that still have high GC content, Fig 2), it is possible that DSB formation can (or at least come close). For example, *Deinococcus radiodurans* is resilient to extremely high rates of DSB formation [60] and has high genomic GC content, but lacks Ku. It is difficult to directly assess the rate of DSB formation, but not impossible [61]. We hope to see future work that assays rates of damage in the environment. DNA damage is an important challenge for microbes to overcome, and a systematic understanding of how damage varies between environments is of broad ecological interest.

## Methods

### Data

We downloaded all available completely sequenced prokaryotic genomes from NCBI's non-redundant RefSeq database FTP site on December 23, 2017 [62] and searched for the presence of the gene coding for the Ku protein, which is central to the NHEJ pathway, using hmmsearch (E-value cutoff $10^{-2}$/#Genomes, Pfam family PF02735; [63]). This identified 21389 genomes containing Ku out of a total 104297 genomes analyzed.

For phylogenetic analyses, we downloaded the SILVA Living Tree 16S rRNA tree [64]. We obtained mutational bias estimates from Long et al. ($m$; [6]), of which 22 could be matched with a genome in our dataset. We obtained estimates of the rate of homologous recombination from [43, 44]. For our analyses on the fate of polymorphism and for estimating recombination we downloaded alignments from the Alignable Tight Genomic Cluster (ATGC) database [42].

Trait data were obtained from the ProTraits microbial trait database (2679 species; [39]). ProTraits scores are expressed as separate confidences that a particular species does or does not have a trait based on the results of an automated text-mining algorithm, giving two scores per binary trait. We combined these scores to obtain what is essentially a probability that a microbe has a given trait using the equation in Weissman et al. [65], yielding a single score between zero and one for each trait for each species. We selected a suite of traits known to be associated with either the incidence of Ku or genomic GC content (Soil-dwelling, aerobicity, growth temperature, nitrogen fixation, spore formation) to include in our analysis. To assess trait vs. Ku relationships we sampled a single genome per species from our RefSeq dataset to determine Ku presence/absence in species with trait data available (617, 2062 without). Most species either always have or always lack Ku (S11 Fig), meaning that sampling should give a reliable estimate of whether we can expect a species to typically have Ku.

We downloaded the complete list of genomes from the REBASE restriction enzyme database [59], which includes all RM enzymes found on a given genome, along with their target sequence if known. Using the listed accession numbers, we then downloaded each corresponding genome from RefSeq in order to assess GC content near restriction sites on the genome (potential sites where DSBs would occur).

Please visit https://github.com/jlw-ecoevo/gcku/ for code and intermediate datasets.

## Phylogenetic linear models

Using the 6648 organisms on the SILVA tree for which we had a genome and could assess Ku presence/absence (2051 with, 4597 without) we built a series of phylogenetically corrected linear models of genomic GC content using the phylolm package in R [66]. First we logit transformed our GC content (%GC) values

$$GC_l = \log\left(\frac{\%\text{GC}}{1 - \%\text{GC}}\right) \tag{1}$$

so that values were in the range $(-\infty, \infty)$. We then used Ku incidence as a binary predictor along with $\log_{10}$ genome length as a continuous predictor to predict $GC_l$ using phylogenetic regression:

$$\mathbf{y} = \mathbf{X}\beta + \boldsymbol{\varepsilon} \tag{2}$$

where

$$\boldsymbol{\varepsilon} \sim \mathcal{N}(0, \sigma_p^2 \mathbf{V} + \sigma_e^2 \mathbf{I}) \tag{3}$$

so that $\mathbf{V}$ is the phylogenetic covariance matrix and $\sigma_e^2$ is the variance of the measurement error [66]. Brownian motion (BM) models of trait evolution are inappropriate when trait values are bounded. While GC content is theoretically bounded at zero and one, there are no species that approach these bounds, and our logit transforming of the GC content values should ameliorate this issue. Sometimes Ornstein-Uhlenbeck (OU) models of trait evolution are also used in these cases. We found that using an OU model had no qualitative effect on our result (Table 1), though this model had a lower AIC than the Brownian motion model (-10760 versus -10754).

We also applied the above analyses independently to the three best-represented phyla in the dataset, each with >1000 genomes: Actinobacteria (614 with Ku, 446 without), Firmicutes (320 with Ku, 699 without), and Proteobacteria (539 with Ku, 1384 without).

Finally, for our "Uniform Ku" models we excluded all genera from our dataset that had fewer than two genomes with which to assess Ku incidence, and then excluded any genera for which Ku incidence was not uniform (all genomes had Ku or all genomes lacked Ku). We then repeated our above analysis (779 taxa with Ku, 2365 without).

## Ancestral state reconstruction

We performed an ancestral state reconstruction of the presence/absence of Ku in the *Baccilaceae* (S8 Fig). We used the R package corHMM to reconstruct the evolutionary history of this trait on the subtree of the SILVA phylogeny describing the *Baccilaceae* [67]. We allowed for up to two rate classes (for trait evolution) across the tree when building our evolutionary model (rate.cat parameter in function corHMM, otherwise default parameters), but found that a model with a single rate class had a lower AICc (257.3119 vs. 263.8347). Thus we only retained a model using a single rate class across the tree.

## Fate of polymorphism

The ATGC database groups closely-related genomes into "clusters" and provides alignments of their core genes [42]. We downloaded multiple alignments corresponding to clusters in the ATGC database that had at least three genomes. In this way we could, at a minimum, identify polymorphisms between two genomes while using a third, more distantly related genome to polarize these polymorphisms. We restricted our analysis to orthologous genes (COGs) that were present in all members of a cluster. For each genome in a cluster we obtained a set of polymorphisms for that genome by comparing to the most similar genome in that cluster, using the most diverged genome in the cluster to polarize these polymorphisms (assuming that the diverged genome represented the ancestral state, and ignoring cases where neither of the other two genomes matched this "ancestral" allele). Similarity was calculated as the percent identity over the entire aligned core genome provided by ATGC. In order to ensure that polymorphisms were recent and had not yet undergone selection, we discarded genomes that were not within 1% pairwise divergence of any other genome in their respective cluster (calculated over the set of core genes provided in the ATGC alignments). We also discarded pairs of genomes that had fewer than 5 informative sites (either GC→AT or AT→GC) in order to avoid extreme expected GC content estimates. Thus we obtained a set of 1868739 polarized polymorphisms for 1643 pairs of genomes for which GC content could be assessed and compared to the background genomic GC content (Fig 3). Expected GC content was calculated as in Long et al [6].

To obtain expected GC content at fourfold degenerate sites we repeated the above analysis only looking at polymorphisms at fourfold degenerate sites (S6 Fig). There are about a third as many polymorphisms in this dataset (574944) but a similar number of genome pairs are retained (1351).

In order to estimate mutational biases, we assume that recent polymorphisms will not have had a chance to undergo selection (or BGC). This is similar to the intuition underlying the McDonald-Kreitman test for selection [41], and similar analyses have been performed in past work on GC content [4, 5]. Therefore we can obtain an estimate of the expected GC content based on mutational bias, and infer that selection (or BGC) is acting if the realized genomic GC content differs from the expectation. In practice, because we are looking at alignable coding sequence, selection is likely to be strong, and may bias our estimates (S12 Fig). This is further compounded by the fact that genomes in a cluster can still be quite diverged, although we control for this by restricting to genomes within 1% sequence divergence from each other. In any case, the direction of bias will be towards the equilibrium GC content, as estimated via the genomic background. Thus, this test for selection suffers somewhat in that it has an increased probability of false-negatives, but this bias should not cause a false signal of selection to occur.

While our estimates do not perfectly align with those found in mutation accumulation experiments (S12 Fig), we note that even within a genus there can be extremely high heterogeneous values for GC↔AT mutation bias. For example, Long et al. [6] estimate the ratio of the rate of GC→AT mutation to the rate of AT→GC mutation to be 4.5 in Vibrio fischeri but 2.3 in Vibrio cholerae. Similarly, they find values of 6.6 in Staphylococcus epidermidis but 4.6 in Staphylococcus aureus, despite having very similar values for genomic GC content (0.33 vs. 0.32) and GC content at fourfold degenerate sites (0.20 vs 0.19). This implies that closely related organisms may have very different mutational biases, making comparisons between datasets challenging.

## Measuring recombination

We obtained all available alignments of shared genes within each cluster of organisms in the ATGC database ([42]). We then ran the program PhiPack [45] using 10000 permutations to

generate *p*-values for the occurrence of recombination in each cluster-gene pair. To correct for multiple testing we used a Benjamini-Hochberg correction with a false-discovery rate of 5%. Altogether this yielded 52117 genes with significant evidence of recombination out of 438580 cluster-gene pairs with sufficient information to run PhiPack. To obtain GC content for each cluster-gene pair we took the mean GC content across sequences in the relevant alignment. To obtain cluster-wide estimates of GC content and Ku incidence we took the mean across genomes associated with organisms in that cluster (each cluster member in ATGC is associated with a RefSeq genome).

## Restriction sites

We identified all genomes encoding restriction enzymes with known restriction sequences in our dataset using the REBASE database [59]. We then restricted our analyses to genomes encoding enzymes that had low-GC content restriction sequences (AT-rich restriction sequences defined as those with $\geq$ 75% AT, *n* = 214, no genomes had multiple enzymes with AT-rich targets). For each remaining genome we mapped the corresponding restriction sequence to the genome itself to find all potential sites of self-targeting. We then calculated the mean GC content of the sites directly flanking these self-targets across the genome, obtaining a value for average GC content for each distance (1-200bp) from the target for each genome.

In order to generate an adequate null for comparison, for each genome-restriction sequence pair in our dataset we generated a novel restriction sequence. To do this, we took each restriction recognition sequence and randomly permuted it to obtain a new sequence with identical base composition (if the permuted sequence was identical to the original, we continued drawing until a different sequence was obtained). We then repeated the above flank-analysis with this set of "fake" restriction recognition sequences (a single, large simulated dataset was generated with 15923 genome-enzyme pairs).

Finally, for each flank-distance (1-200bp) in each genome we calculated the difference in mean GC content of the bases flanking true restriction sites from bases flanking the null sites. We bootstrapped the mean of this distribution for each flank distance across genomes to obtain 95% confidence intervals (Fig 5).

## Supporting information

**S1 Table. Output of linear model relating GC content to environmental variables.** The formal model was GC = $\beta_0$ + $\beta_{Ku}$Ku + $\sum_i \beta_i$trait$_i$ + $\epsilon$, where GC is genomic GC content and Ku is a binary variable representing the presence/absence of Ku.
(PDF)

**S1 Fig. The pairwise correlation between traits among species in the trait dataset.** Note that some traits are highly correlated.
(PDF)

**S2 Fig. The correlation of trait values for microbial species with their average genomic GC content is similar to the correlation of trait values with the presence/absence of Ku.** Note that each point is an individual trait, as shown in Fig 1. The dashed diagonal line indicates the *x* = *y* line. For a direct analysis of the relationship between GC content and Ku incidence among organisms see Fig 2 and Table 1.
(PDF)

**S3 Fig. GC content at fourfold degenerate sites follows a similar pattern to that of genomic GC content overall (Fig 2).** The effect of Ku is significant even taking phylogeny into account

using an identical approach to overall genomic GC content (Table 1).
(PDF)

**S4 Fig. While there is a positive GC content versus genome length trend, genomes with Ku have elevated GC independent of this relationship.** (a) Regression and contour lines were created using default ggplot settings (b) The positive GC versus Ku relationship hold across taxa, independently of any relationship with genome length. Regressions of GC versus log genome length for Ku and non-Ku genomes shown.
(PDF)

**S5 Fig. Mutational bias does not appear to be associated with the NHEJ pathway.** Organisms with the Ku protein did not differ significantly in their GC↔AT mutational biases from those without the Ku protein (t-test, $p > 0.34$). Estimates of mutational bias were obtained from Long et al. [6].
(PDF)

**S6 Fig. Genomes with Ku appear to fix GC alleles at a greater rate than expected (either due to BGC or selection).** (a,b) Genomes with Ku have, on average, even greater elevation of GC over expectation than genomes without Ku. Expected GC estimated from polymorphism data; in contrast to main text Fig 3, here we only use polymorphisms at fourfold degenerate sites. This signal is conservative due to observed polymorphisms experiencing some effects of BGC/selection (see Methods for discussion).
(PDF)

**S7 Fig. Genomes with Ku appear to fix GC alleles at a greater rate than expected (either due to BGC to selection).** Genomes with Ku have, on average, even greater elevation of GC over expectation than genomes without Ku. This figure is identical to panels from Fig 3 and S6 Fig except that we draw loess smoothing lines using default ggplot settings instead of linear model fits.
(PDF)

**S8 Fig. Phylogeny of the *Baccilaceae* (subtree of the SILVA tree).** (a) Ku presence/absence plotted on the tips of the tree as in Fig 2 (blue with, red without Ku). (b) Ancestral state reconstruction of Ku (one rate class). Each internal node is represented by a pie chart describing the probability that that organism either had (black) or did not have (white) Ku. Notice that the root and most nodes near the root are likely to have had Ku.
(PDF)

**S9 Fig. Frequency of Ku presence does not appear to be positively associated with rates of homologous recombination for a species.** (a) Estimated rate of recombination relative to mutation rate from Vos and Didelot [43]. (b,c) Estimated number of recombination events per gene family for species estimated with two methods by Rendueles et al. [44]. In general all of these methods give highly correlated results [44].
(PDF)

**S10 Fig. No relationship between genome-wide recombination frequency and (a) Ku incidence or (b) GC content in the ATGC database.** We used the PHI statistic (see Methods) to determine if genes within each ATGC cluster of genomes had evidence for recombination. The percent of genes with evidence for recombination (out of all genes with sufficient data to test) showed no relationship to either Ku or GC content (averaged across genomes in a particular ATGC cluster).
(PDF)

**S11 Fig. Most species in RefSeq tend to always encode or always lack Ku on their genomes.** Shown is the proportion of genomes within a species that have Ku (all RefSeq assemblies) plotted against the total number of assemblies in RefSeq for that species.
(PDF)

**S12 Fig. We evaluate the use of polymorphisms as a proxy for mutation by comparing estimates for the few species present in both the polymorphism and mutation accumulation data.** (a) Estimates based on all polymorphisms. (b) Estimates based on polymorphisms at fourfold degenerate sites. Here we see selection/BGC appears to bias the polymorphism estimates when mutation is extremely biased towards AT.
(PDF)

## Author Contributions

**Conceptualization:** JL Weissman, Philip L. F. Johnson.

**Formal analysis:** JL Weissman.

**Investigation:** JL Weissman.

**Methodology:** JL Weissman.

**Supervision:** Philip L. F. Johnson.

**Writing – original draft:** JL Weissman, Philip L. F. Johnson.

**Writing – review & editing:** JL Weissman, William F. Fagan, Philip L. F. Johnson.

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
