## [Decision Letter · Decision Letter 0]

1 Oct 2019

Dear Dr Johnson,

Thank you very much for submitting your Research Article entitled 'Linking high GC content to the repair of double strand breaks in prokaryotic genomes' to PLOS Genetics. Your manuscript was fully evaluated at the editorial level and by independent peer reviewers. The reviewers appreciated the attention to an important topic but identified some aspects of the manuscript that should be improved. In particular, reviewer 3 makes suggestions for inclusion of additional data on homologous recombination, and use of the PHI test as in previous comparable studies, and we would like to encourage the authors to follow these suggestions.

We therefore ask you to modify the manuscript according to the review recommendations before we can consider your manuscript for acceptance. Your revisions should address the specific points made by each reviewer.

[LINK]

Yours sincerely,

Xavier Didelot

Associate Editor

PLOS Genetics

Lotte Søgaard-Andersen

Section Editor: Prokaryotic Genetics

PLOS Genetics

Reviewer's Responses to Questions

**Comments to the Authors:**

Reviewer #1: In this study, Weissman and colleagues explore a new mechanism potentially explaining the evolution of GC-content in bacteria. Many works have been published on this question, but yet, no single explanation has imposed itself. The authors hypothesize that DNA breaks and repair could be the underlying cause of GC-content evolution across bacteria. Although they cannot provide direct evidence supporting their hypothesis, the results show, at the very least, a clear link between NHEJ and GC-content. The methods used by the author are generally sound and only have a few criticisms. The manuscript is well written and very pleasant to read. I think this is an interesting hypothesis and a good study. As mentioned by the authors, future experimental works could potentially test this hypothesis.

The methods appear adequate as far as I can judge, and I don’t have any major concerns. However, I am not sure to understand how the authors calculated the incidence of Ku across the data set. For what I understood, it represents the probability that Ku is really present in a genome. Figure S1 is particularly intriguing, but that I am not sure what to get from it. It seems to represent a correlation of correlation coefficients, which might be a very indirect way to show a correlation. It would be more straightforward to represent the correlation between Ku incidence and GC-content directly.

The main issue with the results of the manuscript is that the authors are using the presence of Ku as evidence for more frequent DSBs. As stated by the authors, the NHEJ pathway is either present or absent but DSBs can occur at different rates. Figure 2A is rather convincing but it would be interesting to indicate the size of each sample. Also, it is not clear to me whether the data were computed on the entire dataset of genomes or if only one genome were selected for each species. Species that don’t encode Ku appear to present a wider range of GC-content while species that encode Ku appear much more biased toward high GC-content. It would be informative to explore and discuss the rare species that encode Ku but present a relatively low GC-content. These cases might be insightful, but maybe the authors did not find anything worth reporting in the manuscript.

The authors argue the restriction systems might elevate the frequency of DSBs and use the presence of RM systems as an indirect indicator for elevated DSBs. If we follow their logic, we might expect species encoding larger numbers of RM systems to present higher GC-content. I find this argument not very convincing considering that Helicobacter pylori encodes an exceptionally high number of RM systems (Oliveira, Touchon and Rocha, NAR 2014) but present a relatively low GC-content (~39%).

Finally, I think it is interesting that GC-content and genome length correlate. This observation is not new, but it supports the hypothesis of the authors. I believe it can be safely assumed that, overall, bacteria with larger genomes endure more frequent DSBs. Under the authors’ assumption, the higher GC-content of larger genomes could be explained by the need to repair more frequent DNA breaks. I think this was not explicitly formulated in the manuscript and could be emphasized.

Reviewer #2: I have reviewed this article before for another journal. In this new version, most of my initial critics have been addressed and I find the article quite good. The relationships between genomic GC-content and the presence of the NHEJ pathway is an interesting point to bring to the debate on the evolution of GC-content in genomes. However, I still have difficulties with the hypothesis of the authors that there is selection for high GC-content to favor double strand breaks repair. It is not clear to me why the hypothesis that NHEJ is itself a repair mechanism that is biased toward GC (just like BGC in HR) is not considered and discussed. The argument that GC-rich regions are better repaired seems relatively weak to me.

Reviewer #3: This manuscript presents new observations and a new hypothesis to explain the long-time puzzle of prokaryotic GC content heterogeneity, and the discrepancy between observed GC contents and their – almost universally lower – expected value based on mutational patterns. They report that the non-homologous end-joining (NHEJ) protein Ku is strikingly associated with high genome GC content, and also with the departures from mutational equilibrium, in a stronger way than any previously considered trait (notably those associated to lifestyle). The authors interpret this Ku-GC association as a signature of GC elevation being a response to frequent exposure to double-stranded DNA (dsDNA) break (DSBs). This is considered under several hypotheses, including that GC elevation and Ku occurrence may both be correlated responses the high incidence of DSBs, via separate mechanisms. Alternatively, they investigate a hypothesis where Ku is causally linked to GC elevation, via selective process promoting the elevation of GC content in the genome and in particular in regions susceptible to regular DSBs such as self-target sites for restriction enzymes to improve the efficiency of Ku repair function. They conclude that Ku (or the NHEJ pathway) is unlikely to account on its own for the whole higher-than-expected-GC phenomenon, but may at least be the functional mechanism of a selective process that accounts for part of this phenomenon.

The manuscript is very well written and documented, and presents relevant analyses to test the new hypothesis. The authors also attempt to link these new results to observations made previously regarding other hypotheses of mechanism for above-mutational-equilibrium genome GC contents, namely selection for higher %GC per se, and biased gene conversion (BGC).

The evidence presented in support of the Ku-GC association is sufficient and convincing, and its interpretation is cautiously discussed to consider known evidence, and to take into account potential interactions or confounding signatures with other mechanisms.

However, it would be desirable that the authors bring their study a step further, and bring a bit more material to help the reader (and future investigations) to resolve this puzzle. Namely, in order to test the relevance of the BGC hypothesis in the light of the facts presented in this study, they confront Ku occurrence and GC content data are to homologous recombination (HR) data, which are only recovered from other studies. This brings the concern that these data are not properly matched with the sty’s own datasets: summary statistic from studies using different genome sets (and potentially different set of sequences within genomes) on the basis of the sole species name is unlikely to reflect the exact properties of the genome datasets investigated here. Considering the scale of the present investigation (the whole prokaryotic tree of life), it is crucial that each data point be accurately representing the properties of the considered organism, and hence that all measurements be made on the same dataset. As explained below, applying the HR test/quantification procedures described in the cited literature to this dataset would be a feasible undertaking, and would add much value to the paper.

Finally, I notice that the intermediary data is not made available. This includes tables describing the sets of genomes used, the occurrence of Ku in these genomes, the list of restriction enzyme found in them and their corresponding target sequences, the genome tree presented Fig 2 in machine-readable format, the estimates of GC at the mutational equilibrium, etc. the scripts used to generate such data, as well as those used to test their association, should be provided as well. I think that publication in scientific journal, and especially in the open-access pioneer PLoS journals, should always be backed by full access to data and proceedings of the analyses so they can be replicated. Please attach them as a supplement, or provide a link to an external data/code repository (my recommendation).

I let the editor appreciate the relevance of the request for additional data on HR. Provided that the few minor comments below are addressed and that intermediary data are provided, I think the manuscript would be otherwise generally fit for publication in PLoS Genetics. I thus recommend the paper for minor revision.

Detailed comments

L70-82: this paragraph belongs to the introduction, with which it is slightly redundant.

L87-89: this correlation of the Ku and GC, as revealed by correlation of each with ‘third-party’ trait, is striking. However, it would be nice to have a more straightforward estimate and visualisation of their association. Could the authors provide a correlation r^2 and p-value for GC ~ Ku occurrence? In complement of the PCA in fig. 1, could they also plot the result of a linear discriminant analysis (LDA) maximizing the separation of the samples based on their Ku +/- state, and plotting the %GC over it (as well as showing the explained variance of such a projection)?

Actually, something like a heatmap of a correlation matrix of all these traits would be helpful (in supplement) for the reader to see how the traits are associated with each other.

L90-92 / S1 Table: I think that the table legend should spell out how the model was formulated (like give the R code or a more formal string like ‘y ~ trait1 + trait2’). Once that is clarified, it would be interesting to present results of a general linear model where the prioritization of would have been different: with Ku as first explanatory variable, would the other traits have any variance left to explain?

L95-96: “In fact this is trivially true, as Ku presence is a discrete, binary variable whereas the rate of DSB formation is continuous.”

This is a relevant point, and should be considered further. In fact, the presence/absence of the Ku protein (used as a proxy of a functional NHEJ pathway) is a trait that can vary among strains of a clade or species, as stated by the authors L176-178. Transitions between the Ku +/- states might have happened recently in certain strain lineages, and at potentially high frequency over time. On the contrary, %GC increase is expected to be a long process, given that the effect size of either selection for higher %GC or BGC phenomena are likely small, that they act against the mutational bias, and that selection for other traits may interfere with this background amelioration process. This is to be opposed to phenotypic traits (usually considered for correlation under BM or OU models) that result from the expression of the genotype of an individual organism, i.e. in sync with its current genotype.

It follows that the association between a potentially recently acquired trait (Ku presence) and the result of a long-standing process (%GC increase away from the mutational equilibrium) could possibly be coincidental. The authors should try and repeat their analyses by restricting them to genomes in clades where the Ku +/- state is conserved, and where we can expect that it has been present/absent for long enough so that the base substitution process is in its steady state. The situation that “Ku presence/absence is sprinkled throughout the prokaryotic phylogeny”, and described in Figure 2B, where it seems that many clades have a homogeneous pattern of Ku occurrence, should allow them to run such restricted analyses with enough statistical power (while still using the phylogenetically-aware regression models to avoid over-counting the replicated data points within such homogeneous clades).

This is an important point, as most studies trying to confirm/invalidate the hypothesis of BGC have tried to correlate the %GC with the recombination rate inferred from recent polymorphism data, which again reflect a recent property of the population, but might not reflect the long-term average recombination rate that the lineage has experienced – a major issue that prevented most past analyses to settle the debate on the existence or not of BGC in Prokaryotes. Ku occurrence is a simple binary trait and its past distribution is more easily estimated than the past recombination rate, which estimation from polymorphism data is inherently biased towards recent times due to saturation of homoplasy signals; by studying this simpler trait, the authors here have an opportunity to bring stronger evidence on that subject than any other previous study.

Section “No Apparent Relationship Between Rate of Homologous Recombination and NHEJ”:

I agree with the general conclusions of the authors for this section, that is the impossibility to conclude given the data, but I think they could try and provide further evidence to fuel the debate. In particular, they only rely on data from previous study to quantify the effect of homologous recombination (HR) on species they investigated in their own dataset. The third-party data they report is likely to be inadequate to answer the question asked, for several reasons.

The quantification of HR rates (r/m) by Vos and Didelot (ref [44]) is made using ClonalFrame, a method that is able to grasp the long-term average HR rate (see comment above), which is a good thing, but was based on multi-locus data and on quite a variable set of strains depending on the species, thus unlikely to reflect findings from sets of whole-genomes of calibrated diversity (from the ATGC database) used in the present study.

The data from Ruendules et al. [45] are also unlikely to have used the same set of genomes, and use simple linkage disequilibrium-based metrics which have been designed to perform test of occurrence of HR, not to quantify it, and which application at the whole-genome scale is unlikely to grasp any nuance in such signal.

The fairer comparison is with the data from Lassalle et al. (ref [7]), but again the genome datasets are unlikely to be matched. Published genome data expand rapidly and, as a consequence, prokaryotic species definitions are being regularly revised; the genomes available for what was considered to be B. anthracis by Lassalle et al. in 2015 is thus unlikely what is available today in ATGC database under this same name. I believe this drastically limits the scope of what the authors are able to say about HR in the framework of this study.

I would suggest that the authors replicate the procedure used by Lassalle et al., that is running the PHI test on the core gene alignments of their species datasets (or at least a representative subset), as provided by the ATGC database. The PHI test is very fast and can easily be ran in parallel on a large collection of gene alignments. This is not essential to the core argument of the paper, but would help going further on the matter.

L216-220: “the extremely strong and specific association between GC and Ku suggests that this relationship may be particular to the specific conditions selecting for Ku (especially considering the absence of an association between HR and GC when looking between genomes [5]; S7 Fig)”

As discussed above, these datasets are very unlikely to be matched with the authors’, and rejecting the association of elevated %GC (or Ku occurrence) with HR rates on this basis is possibly flawed. Again, I would suggest the authors run their own recombination tests/quantifications on their own datasets so they can draw robust conclusions.

L217 “association between GC and Ku”; L219 “association between HR and GC”; L223 “association 223 between NHEJ and high GC content” and more:

The authors need to use a consistent term to refer to the A/T vs. G/C base composition of genomes; the early sections of the manuscript use the acronym ‘%GC’, but later just name it ‘GC’, or ‘GC content’. One term should be chosen and used throughout the manuscript

L223: “Given our lack of enthusiasm for BGC as a mechanism”

I appreciate the author’s willingness to disclose any subjective bias they may have towards one or another scientific hypothesis, but I don’t think it is appropriate to use it to justify what they investigate. Please rephrase into something like “Given the lack of evidence in support of the BGC hypothesis as reported above, we chose to investigate an alternative hypothesis.”

Importantly, the authors should make clear that they are not opposing hypotheses, i.e. rejecting BGC because of support for the selection hypothesis, or vice versa. In principle, both hypotheses could be true, and so could be a third (or more) alternative that was not yet proposed in the literature.

L228-229: “high GC content may promote DNA repair, both by facilitating canonical NHEJ 228 (i.e., Ku-dependent) and alternative NHEJ (i.e., Ku-independent) pathways.”

Please cite relevant literature supporting these claims. If they are supported by the references [50, 51, 52] cited in the following paragraph, please connect these text sections (e.g. by not ending the sentence L229 and connecting it to the next with a colon) so to make it clear.

L232-234 “Any factor that stabilizes this interaction (e.g., high GC via an increased number of hydrogen bonds) may thus increase the efficiency of NHEJ repair.”

L238-239: “It stands to reason that high GC content in these regions of microhomology might help stabilize the end-pairings and improve the efficiency of repair.”

Again, please cite the relevant literature (redundancy of citation with the previous sentence is not an issue in my opinion) so to clarify whether this is a (reasonable) speculation of mechanism by the authors or something that is backed by experimental evidence.

L235-237: “alternative high-fidelity end-joining pathways that are independent of the NHEJ machinery, and that these pathways are primarily dependent on nearby microhomology to tether the DNA ends together”

Please clarify which bits of sequence are required to present microhomology for the NHEJ or NHEJ-independent end-joining pathways to function. If it is the immediate sequence on both free ends of the broken dsDNA, this means that sequences with short repeats would be more likely to be repaired by these pathways. This would come as a confounding factor for the prediction of effect of %GC in this system (for instance, because short repeats are enriched in mobile elements like phages, transposons or integrons, which are themselves generally AT-rich…); the authors should mention these potential pitfalls as they develop this hypothesis.

L262-264: “We further predicted that, for restriction enzymes with low GC recognition sequences, the bases flanking restriction sites on the genome would have elevated GC”

The test presented afterwards could also support selection-free hypotheses where the converse rationale would stand, i.e. that increased repair at those DSB-prone sites would induce higher %GC; typically, it would be in line with the BGC hypothesis as HR-associated pathways are also taking part in the repair or restriction enzyme-induced breaks.

L259: “to help ameliorate the effects of autoimmunity”

‘mitigate’ instead of ‘ameliorate’

L316:” This identified 21389 genomes containing Ku out of a 316 total 104297 genomes analysed”

Please provide a list of the genomes, and of which were deemed positive for the Ku protein-coding gene.

L322: “we downloaded alignments from the Alignable Tight Genomic Cluster (ATGC) database [43]”

Please provide the list of genomes assigned to cluster, the number of gene alignments and clarify how many were dropped/retained when filter were applied.

L324: “Trait data were obtained from the ProTraits microbial trait database (2679 species; [39])”

Please provide the table of how genomes from RefSeq were matched with entries of ProTraits (or if sharing identifiers, a list of genomes covered by both databases).

L382: “The rationale behind this test”

No test has been described at this point; I assume the authors refer to the comparison of the expected %GC (based on the mutational pattern estimated from phased polymorphism data) to the realized genomic %GC, which they describe right after; please rephrase.

L413: “no genomes with multiple AT-rich enzymes”

Please clarify how you define AT-rich enzymes (if based on the composition of the target sequence, what threshold of %GC?).

L424: “We then repeated the above”

Please specify how many draws of thee permutations were conducted.

**Have all data underlying the figures and results presented in the manuscript been provided?**

Reviewer #1: Yes

Reviewer #2: No: A link to a table including all the features of each genome (GC, genome size, ecological information, taxonomy, gene presence, etc...) should be provided to allow reproducibility of the results.

Reviewer #3: No: the intermediary data is not made available. This includes tables describing the sets of genomes used, the occurrence of Ku in these genomes, the list of restriction enzyme found in them and their corresponding target sequences, the genome tree presented Fig 2 in machine-readable format, the estimates of GC at the mutational equilibrium, etc. the scripts used to generate such data, as well as those used to test their association, should be provided as well

PLOS authors have the option to publish the peer review history of their article (what does this mean?). If published, this will include your full peer review and any attached files.

Reviewer #1: No

Reviewer #2: No

Reviewer #3: Yes: Florent Lassalle

---

## [Editor Report · Decision Letter 1]

25 Oct 2019

Dear Dr Johnson,

We are pleased to inform you that your manuscript entitled "Linking high GC content to the repair of double strand breaks in prokaryotic genomes" has been editorially accepted for publication in PLOS Genetics. Congratulations!

Yours sincerely,

Xavier Didelot

Associate Editor

PLOS Genetics

Lotte Søgaard-Andersen

Section Editor: Prokaryotic Genetics

PLOS Genetics

Comments from the reviewers (if applicable):

**Data Deposition**

http://datadryad.org/submit?journalID=pgenetics&manu=PGENETICS-D-19-01378R1

**Press Queries**

---

## [Editor Report · Acceptance letter]

1 Nov 2019

PGENETICS-D-19-01378R1 

Linking high GC content to the repair of double strand breaks in prokaryotic genomes 

Dear Dr Johnson, 

We are pleased to inform you that your manuscript entitled "Linking high GC content to the repair of double strand breaks in prokaryotic genomes" has been formally accepted for publication in PLOS Genetics! Your manuscript is now with our production department and you will be notified of the publication date in due course.

With kind regards,

Matt Lyles

PLOS Genetics

On behalf of:
